# Tadalafil Treatment of Mice with Fetal Growth Restriction and Preeclampsia Improves Placental mTOR Signaling

**DOI:** 10.3390/ijms23031474

**Published:** 2022-01-27

**Authors:** Kayo Tanaka, Hiroaki Tanaka, Ryota Tachibana, Kento Yoshikawa, Takuya Kawamura, Sho Takakura, Hiroki Takeuchi, Tomoaki Ikeda

**Affiliations:** Department of Obstetrics and Gynecology, Mie University School of Medicine, Edobashi, Tsu 5148507, Mie, Japan; tanaka-ky@clin.medic.mie-u.ac.jp (K.T.); r-tachibana@clin.medic.mie-u.ac.jp (R.T.); yoshikawa-k@clin.medic.mie-u.ac.jp (K.Y.); kwmrtky1112@yahoo.co.jp (T.K.); s-takakura@clin.medic.mie-u.ac.jp (S.T.); h-takeuchi@clin.medic.mie-u.ac.jp (H.T.); t-ikeda@clin.medic.mie-u.ac.jp (T.I.)

**Keywords:** fetal growth restriction, placenta, mTOR signaling, tadalafil, preeclampsia, L-NAME

## Abstract

Fetal growth restriction (FGR) is a major cause of poor perinatal outcomes. Although several studies have been conducted to improve the prognosis of FGR in infants, no effective intrauterine treatment method has been established. This study aimed to use tadalafil, a phosphodiesterase 5 inhibitor (PDE5) inhibitor, as a novel intrauterine treatment and conducted several basic and clinical studies. The study investigated the effects of tadalafil on placental mTOR signaling. Tadalafil was administered to mice with L-NG-nitroarginine methyl ester (L-NAME)-induced FGR and associated preeclampsia (PE). Placental phosphorylated mTOR (p-mTOR) signaling was assessed by fluorescent immunohistochemical staining and Western blotting. The expression of p-mTOR was significantly decreased in mice with FGR on 13 days post coitum (d.p.c.) but recovered to the same level as that of the control on 17 d.p.c. following tadalafil treatment. The results were similar for 4E-binding protein 1 (4E-BP1) and S6 ribosomal (S6R) protein, which act downstream in the mTOR signaling pathway. We demonstrate that the tadalafil treatment of FGR in mice improved placental mTOR signaling to facilitate fetal growth. Our study provides the key mechanistic detail about the mode of action of tadalafil and thus would be helpful for future clinical studies on FGR.

## 1. Introduction

Fetal growth restriction (FGR) is a major cause of poor perinatal outcomes [1,2]. At present, there is no effective intrauterine treatment for FGR, and children with FGR not only have a high perinatal mortality rate but also remain at risk for poor neurological prognosis even if they survive [3,4]. Although many studies have been conducted to improve the prognosis of FGR in infants, no effective intrauterine treatment method has been established. Moreover, studies have explored when to terminate a pregnancy; however, no standardized method to improve the prognosis has been identified [5,6]. 

The efficiency of tadalafil, a phosphodiesterase 5 inhibitor (PDE5) inhibitor, to treat FGR is evident [7,8,9,10,11]. However, the exact mechanism of its action remains elusive. Therefore, we focused on the mammalian target of rapamycin (mTOR), one of the major regulatory factors associated with fetal growth, particularly mTOR complex1 signaling, to clarify the mechanism of action of tadalafil in the FGR placenta. Tadalafil inhibits PDE5 in smooth muscle cells, thereby inhibiting the degradation of local cGMP, which relaxes smooth muscle cells. This increases uteroplacental blood flow and placental oxygenation, facilitating the growth of the fetus. We have also shown that in mice, tadalafil dilates the maternal vascular sinuses of the placenta and improves hypoxic changes in the placenta [7]

Mechanistic target of rapamycin (mTOR) signaling in the placenta plays an important role in fetal growth and development, and its dysregulation may lead to miscarriage [12]. mTOR, a serine/threonine kinase, is highly expressed in syncytiotrophoblasts of the human placenta [13]. It senses various signals such as growth factors, amino acids, and stress from intracellular and extracellular sources and regulates cell growth, metabolic activity, and survival [14,15,16,17,18]. mTOR forms two intracellular complexes, mTOR complex (mTORC)1 and mTORC2, both expressed in the placenta. mTORC1 promotes protein translation through the phosphorylation of S6 Kinase 1 (S6K1) and 4E-binding protein 1 (4E-BP1) and is mainly involved in regulating cell growth and metabolism [14,15,16,17,18,19], whereas mTORC2 plays an important role in cell proliferation and survival. Furthermore, mTORC1 acts as a placental nutrient sensor indirectly by regulating trophoblast ATP production, which is critical for protein synthesis and energy-intensive active transport in the placenta [20]. Recently, it has been shown that mTOR signaling “directly” regulates nutrient transport from the maternal circulation to the placenta [13,21,22,23,24,25]. Furthermore, several studies based on the phosphorylation of key downstream targets of mTOR have shown that mTOR signaling activity is suppressed in the human FGR placenta [26,27,28,29]. Therefore, we hypothesized that assessing the effects of tadalafil on mTOR signaling activity in the FGR placenta could provide insights into understanding its therapeutic effect on FGR and the underlying mechanism of action. 

The purpose of this study was to evaluate the effect of tadalafil on mTORC1 signaling activity in the FGR placenta. Furthermore, to understand its mechanism of action, we investigated the expression of two key target proteins, 4E-BP1 and S6 ribosomal protein (S6R), the key downstream targets of mTORC1 signaling. This study will improve the key mechanistic detail about the mode of action of the beneficial effects of tadalafil on FGR, and thus will be helpful for future clinical studies on FGR.

## 2. Results

### 2.1. Expression of Phosphorylated mTOR (p-mTOR) in Mice Placenta

#### 2.1.1. Placental p-mTOR Expression on 13 Days Post Coitum (d.p.c.)

To analyze the effect of L-NG-nitroarginine methyl ester (L-NAME)-induced FGR and associated preeclampsia (PE) on p-mTOR signaling in the placenta, we performed fluorescent immunohistochemical staining using anti-phospho-mTOR (Ser2448) antibody on 13 d.p.c. (Figure 1). The percentage area expressing p-mTOR in the labyrinth zone of mouse placenta was 13.8 ± 1.97% in L-NAME mice (L) and 45.1 ± 3.28% in control mice (C), which was significantly lower in Group L (*p* < 0.05).

#### 2.1.2. Placental p-mTOR Expression on 17 d.p.c

Further, we subjected the placenta of mice treated with L-NAME and tadalafil (Group LT) to immuno-histochemical analysis on 17 d.p.c. (Figure 2). The percentage area with p-mTOR expression in the labyrinth zone of mice placenta was 58.5 ± 2.51% in control mice (C), 32.2 ± 3.52% in L-NAME mice (L), and 52.4 ± 3.07% in L-NAME-tadalafil mice (LT). Similar to the results on 13 d.p.c., p-mTOR expression in Group L was significantly lower (*p* < 0.01) than that of Group C, but its expression in Group LT was not significantly different from that of Group C. In addition, the percentage of area expressing p-mTOR was significantly higher in Group LT than in Group L (*p* < 0.05).

### 2.2. Expression Analysis of Phosphorylated 4E (eIF4E)-Binding Protein 1 (p-4E-BP1) and Phosphorylated Ribosomal S6 Protein (p-S6R) in Mouse Placenta

#### 2.2.1. 4E-BP1 and S6R Expression in Placenta on 13 d.p.c.

Next, we investigated the expression of 4E-BP1 and S6R, the downstream signaling proteins of mTORC1, a functional partner of mTOR. The results are shown in Figure 3. The expression level of 4E-BP1 total protein was not significantly different between the two groups (102.2 ± 2.9% in Group C and 109.0 ± 12.1% in Group L); however, the expression of its phosphorylated form, p-4E-BP1 (Thr37/46) was significantly lower in Group L (58.1 ± 8.6%) than that in Group C (81.1 ± 16.4%) (*p* < 0.05).

Similarly, total S6R protein expression was not significantly different between the two groups (95.1 ± 26.0% in Group C and 88.4 ± 31.6% in Group L). However, p-S6R (Ser235/236) expression was significantly lower in Group L (58.1 ± 8.6%) than in Group C (81.1 ± 16.4%) (*p* < 0.05).

#### 2.2.2. 4E-BP1 and S6R Expression in Placenta on 17 d.p.c.

We tested the effect of tadalafil treatment on mTORC1 downstream signaling on 17 d.p.c. As shown in Figure 4, the expression of 4E-BP1 (total) and S6R (total) was not significantly different between the three groups [Groups C (*n* = 5), L (*n* = 5), and LT (*n* = 7)]. However, expression levels of p-4E-BP1 and p-S6R decreased significantly in Group L compared with those in Group C, which was restored by tadalafil treatment in Group LT. The mean ± SE obtained for each dataset was as follows: p-4E-BP1(Thr37/46): 91.6 ± 10.4%, 68.1 ± 15.3%, and 97.0 ± 22.2%; p-4E-BP1(Thr70): 84.5 ± 19.5%, 58.4 ± 9.6, and 75.0 ± 14.1%; p-S6R (Ser235/236): 90.6 ± 18.3%, 56.3 ± 10.7%, and 85.0 ± 24.0% in Groups C, L, and LT, respectively.

## 3. Discussion

In the present study, we investigated mTOR signaling in the placenta of the L-NAME-induced FGR and PE mouse model. In the syncytiotrophoblast of the placenta, mTOR forms mTORC1 in association with several other metabolic signals [30]. mTORC1 regulates metabolism, growth, cell division, and cell death in response to growth factors, such as insulin and insulin-like growth factor-1 (IGF-1), intracellular ATP levels, hypoxia, DNA damage, biomolecules such as amino acids, glucose, and fatty acids. It plays a central role in the regulation of growth, cell division, and cell death. It has also been reported that placental mTORC1 regulates active transport via nutrition transporters and mitochondrial function [20], and its activity has been shown to be suppressed in FGR [26,27,28,29]. These studies indicate that mTOR signaling plays a pivotal role in fetal growth and development. 

This study investigated the effect of an FGR inducer, L-NAME, and FGR suppressor, tadalafil, on mTOR signaling. We further tested the expression of two key target proteins, 4E-BP1 and S6R, which act downstream in the mTORC1 signaling pathway. On 13 d.p.c., L-NAME administration significantly decreased the expression of p-mTOR (Ser2448), p-4E-BP1 (Thr37/47), and p-S6R (Ser235/236) in the mouse placenta, indicating that mTOR signaling activity was suppressed. In contrast, tadalafil treatment restored expression of all the three proteins to the same level as that of the control group on 17 d.p.c. Therefore, we concluded that tadalafil treatment restored mTOR signaling, which was suppressed in the L-NAME-induced FGR placenta. Concordantly, several previous studies have reported the suppression of mTORC1 signaling activity in the FGR mouse model [26,28,31,32]. 

We previously reported that the administration of tadalafil to the L-NAME-induced FGR mouse model facilitates fetal development [7]. Therefore, we used the L-NAME-induced FGR model for studying the effects of tadalafil on mTOR signaling. Our previous study demonstrated that the effect of tadalafil on fetal growth is modulated via improved placental hypoxia [7]. As tadalafil has a vasodilator effect by enhancing cGMP levels in vascular smooth muscle cells, the study inferred that it improves fetal placental blood circulation by dilating uterine arteries, leading to subsequent improvement of hypoxia and mTOR signaling in the placenta of tadalafil-treated mice. However, there is enough evidence to suggest that tadalafil may not act through the uterine artery dilation route alone [9]. For example, in our previous study, tadalafil administered to a reduced uterine perfusion pressure (RUPP) mouse model demonstrated a fetal growth effect [9]. In the RUPP mouse model, PE and FGR were induced in the pregnant mouse by ligating the ovarian and uterine arteriovenous arteries with a nylon thread, which reduces uterine pressure [33,34]. In this RUPP mouse model, although fetal growth was improved, the uterine artery diameter did not change with tadalafil treatment, suggesting the possibility of a direct effect of tadalafil rather than through uterine artery dilation. Furthermore, tadalafil also has ameliorating effects on vascular endothelial damage in addition to anti-inflammatory and antioxidant effects [9,35,36]. Therefore, it can be expected that the improvement in mTOR signaling activity in the present study could also be modulated via some of these direct effects of tadalafil.

In addition to tadalafil, other PDE5 inhibitors include sildenafil and vardenafil, which differ mainly for their time to effect, half-life, and PDE selectivity [37,38,39]. Tadalafil is 187 times more selective for PDE5 than PDE6, whereas it is significantly less selective for PDE11 A (~5 times more selective for PDE5 than for PDE11 A) [40]. Although several clinical studies of sildenafil treatment for FGR have been conducted in several countries, reports on its efficacy are scarce [41,42,43]. Furthermore, tadalafil has been shown to have a different effect on pulmonary hypertension than sildenafil [38], indicating the clinical significance of the effect of tadalafil on FGR as well. Therefore, the results of this study may support future clinical studies.

This study is limited by the fact that it was restricted to a single mouse model of L-NAME-induced FGR and associated PE. In this study, we have shown that tadalafil alters mTOR signaling to facilitate fetal development via enhanced fetal placental circulation in the placenta. However, it is important to elucidate further the detailed mechanism of action of tadalafil and to ascertain its role on mTOR signaling in FGR and PE treatment through clinical studies. 

## 4. Materials and Methods

This study was conducted under the review and approval of the Mie University Animal Experiment Committee (Approval code 273111 and date of approval: 1 May 2020).

### 4.1. Establishment of FGR Mouse Model and Experimental Protocol

We have previously established a mouse model of FGR with PE in which the dams exhibited hypertension and proteinuria, and the fetus exhibited growth restriction when the nitric oxide synthase inhibitor L-NAME (Cayman Chemical, Ann Arbor, MI, USA) was administered to pregnant mothers. FGR was induced in C57BL/6 pregnant mice by administering L-NAME solution at a concentration of 1 mg/mL starting 11 d.p.c. [7] The average dose of L-NAME was 280.9 ± 22.5 mg/kg bodyweight (BW)/day. 

Twenty pregnant C57BL/6 dams (CLEA Japan, Tokyo, Japan) were purchased on 9 d.p.c. and divided into two groups based on their matching weight and blood pressure: a control group (Group C, *n* = 6), which received 0.5% carboxymethyl-cellulose (CMC; Wako Pure Chemical Industries, Osaka, Japan) dissolved in drinking water on 11 d.p.c and L-NAME-treated group (Group L, *n* = 14) received dissolved in 0.5% CMC and 1 mg/mL L-NAME (Cayman Chemical) starting on 11 d.p.c. After confirming that the maternal blood pressure in Group L increased (an indicator of PE), Group L was further divided into two subgroups: one subgroup continued to receive only L-NAME (Group L, *n* = 7), while the other subgroup (Group LT, *n* = 7) received L-NAME with 0.08 mg/mL tadalafil (Cayman Chemical Company) suspended in 0.5% CMC, starting on 13 d.p.c. Tadalafil administration was continued until 17 d.p.c. The average dose of tadalafil was 19.3 ± 1.3 mg/kg BW/day. All dams were sacrificed on 17 d.p.c., and their placentas were harvested for further analysis. 

Another set of dams was prepared following the same procedure described above (Group C, *n* = 4; Group L, *n* = 5) and sacrificed on 13 d.p.c. Their placentas were collected to evaluate the effect of L-NAME administration on mTOR signaling. Placentas for histological evaluation were fixed in 4% paraformaldehyde (Nacalai Tesque, Kyoto, Japan) in 0.2 M sodium phosphate buffer (PBS) (pH 7.4) and then embedded in paraffin (Merck Ltd., Frankfurter, Germany) using standard procedures. Placentas for protein analysis were frozen in liquid nitrogen immediately after collection and stored at −80 °C until further analysis.

### 4.2. Fluorescent Immunohistochemical Staining

The Paraffin blocks were sectioned at a thickness of 5 μm and subjected to immunohistochemical analysis. For the evaluation of phosphorylated mTOR (p-mTOR), an anti- phospho-mTOR (Ser2448) antibody (1:100, #ab109268, Abcam, Cambridge, UK) was used as the primary antibody, and 4’,6-diamidino-2-phenylindole (DAPI) was used for nuclear staining. Sections were incubated overnight at room temperature with the primary antibody, followed by an appropriate secondary antibody for 2 h at room temperature. Fluorescence images taken by FV1000-D IX81 confocal laser microscope (Olympus, Tokyo, Japan) were analyzed using Image J software (Wayne Rasband, National Institutes of Health, Bethesda, MD, USA) to determine the percentage of p-mTOR-positive area in the labyrinth zone of the mouse placenta.

### 4.3. Western Blotting

Frozen mouse placenta was washed with PBS, and protein was extracted using radioimmunoprecipitation assay (RIPA) buffer (Nacalai Tesque, Kyoto, Japan). Bicinchoninic acid (BCA) assay was performed to quantify the isolated protein quantification using the Protein Assay BCA kit (Nacalai Tesque). Each sample containing 20 μg of total protein was loaded into a well of polyacrylamide gel (ATTO, Tokyo, Japan) and subjected to Western blotting using standard procedures. Levels of 4E-BP1 and S6R were analyzed to assess the tadalafil effect on the downstream signaling of mTOR. For phosphorylation studies, we used Thr37/46 (1:5000, #9459, Cell Signaling Technology, Danvers, MA, USA) and Thr70 (1:5000, #9455, Cell Signaling Technology) for 4E-BP1 protein and Ser235/236 (1:5000, #2211, Cell Signaling Technology) for S6R protein. For total (unphosphorylated and phosphorylated) protein detection of 4E-BP1 and S6R, we used 4E-BP1 antibody (1:5000, #9452, Cell Signaling Technology) and S6R antibody (1:5000, #2217, Cell Signaling Technology). Chemiluminescence detection was performed using HRP-conjugated secondary antibodies, and the developed blots were quantified using ImageJ software. β-actin (1:2000, #sc-47778, Santa Cruz Biotechnology, Dallas, TX, USA) was used as a loading control for data normalization.

### 4.4. Statistical Analyses

All values are presented as mean  ±  SE Statistical analyses were performed using GraphPad Prism7 (GraphPad, San Diego, CA, USA). The Student’s t-test was used for two-group comparisons, and one-way ANOVA was used for three-group comparisons. A value of *p* < 0.05 was considered statistically significant.

## 5. Conclusions

Tadalafil treatment improved placental mTOR signaling in L-NAME-induced FGR and PE mouse model. This study elucidates the first mechanistic insight into how tadalafil facilitates fetal growth in FGR mice and provides an important framework for future studies.

## Figures and Tables

**Figure 1 ijms-23-01474-f001:**
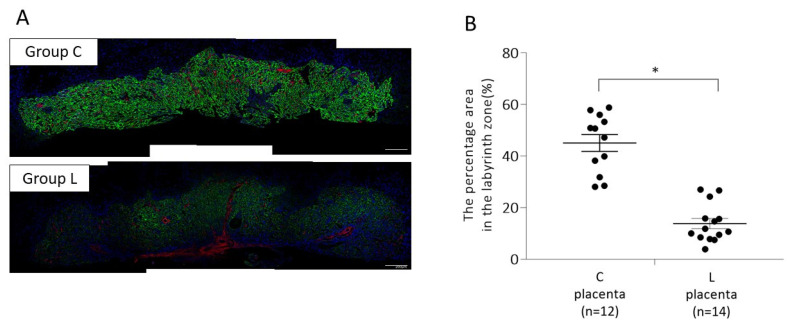
Expression of phosphorylated mechanistic target of rapamycin (p-mTOR) (Ser2448) in the placentas on 13 d.p.c. Paraformaldehyde-fixed, paraffin-embedded placental tissues were used to confirm the expression of p-mTOR by fluorescent immunohistochemical staining. (**A**) Representative images of p-mTOR (Ser2448) expression (green) in group C (upper panel) and group L (lower panel). (**B**) The percentage of p-mTOR (Ser2448)-positive area in control and L-NAME placenta. Data are expressed as the mean ± standard error (SE). Group C, control mice; Group L, L-NAME-induced FGR mice. * *p* < 0.05.

**Figure 2 ijms-23-01474-f002:**
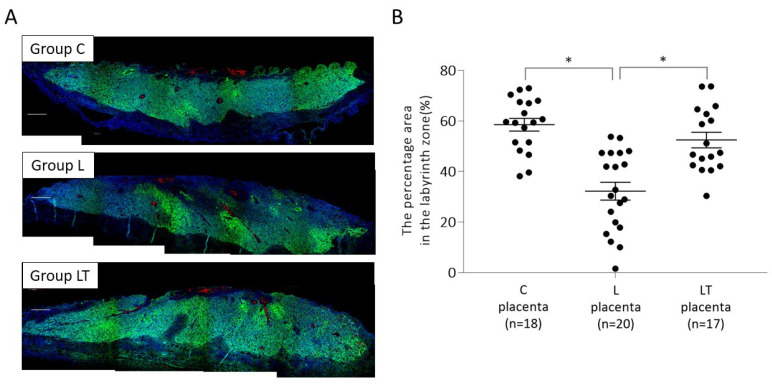
Expression of p-mTOR (Ser2448) in the placentas on 17 d.p.c. Paraformaldehyde-fixed, paraffin-embedded placental tissues were used to confirm the expression of p-mTOR by fluorescent immunohistochemical staining. (**A**) Representative images of p-mTOR (Ser2448) expression (shown in green) in group C (upper panel), group L (middle panel), and group LT (lower panel). (**B**) The percentage of p-mTOR (Ser2448)-positive area in three groups. Data are expressed as mean ± SE. Group C, control mice; Group L, L-NAME-induced FGR mice without tadalafil treatment; Group LT, L-NAME-induced FGR mice with tadalafil treatment. * *p* < 0.05.

**Figure 3 ijms-23-01474-f003:**
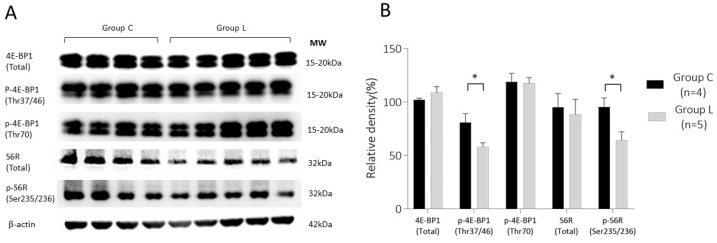
Expression analysis of key downstream targets of mTOR complex 1 (mTORC1) signaling; 4E-BP1 and S6R in the placenta on 13 d.p.c. (**A**) Representative Western blot images of each target. (**B**) Comparison of relative density between Group C and Group L. Data represent the mean ± SE. Group C, control mice; Group L, L-NAME-induced FGR mice; 4E-BP1, eIF4E-binding protein 1; S6R, ribosomal S6 protein; MW, molecular weight. * *p* < 0.05.

**Figure 4 ijms-23-01474-f004:**
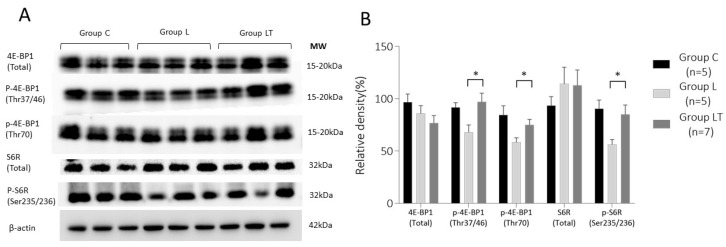
Expression analysis of key downstream targets of mTORC1 signaling; 4E-BP1 and S6R in the placenta on 17 d.p.c. (**A**) Representative Western blot images of each target. (**B**) Comparison of relative density among the three groups (Groups C, L, and LT). Data represent the mean ± SE. Group C, control mice; Group L, L-NAME-induced FGR mice; 4E-BP1, eIF4E-binding protein 1; S6R, ribosomal S6 protein; MW, molecular weight. * *p* < 0.05.

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
