# Peer review of "Tadalafil Treatment of Mice with Fetal Growth Restriction and Preeclampsia Improves Placental mTOR Signaling"

_ijms, 2022, doi:10.3390/ijms23031474_

Round 1

Reviewer 1 Report

The paper presents the results of experimental in vivo studies on influence of tadalafil on the activation of mTOR in fetal in mice. The Authors emphasized that proposed manuscript continues their study on the identification of molecular mechanism of tadalafil positive effect on fetal growth in an FGR mouse model. The manuscript presents results interesting for readers, however there are some details which need to be explain more specific:

  • In the Introduction present more detailed information about significance of mTOR and its signaling targets on fetal growth and development;
  • present more details about chosen dose of studied compounds;
  • describe the number of animals in each group studied;
  • In case of western blots please add the molecular weight of detected protein bands; explain why in the immunoblots (Figure 3, Figure 4) are visible two bands and which of them represents p4E-BP-1 (Thr 37/46) or p4E-BP-1 (Thr70); please, add the band of unphosphorylated 4E-BP-1 protein, as well as for S6R – as a control of changes resulting by phosphorylation activity; was the level of mTOR protein determined?

Author Response

Point-by-point responses to the comments of the Reviewers

Dear Reviewer,

Thank you for your efforts to review our manuscript. Thank you very much for your helpful comments and suggestions. Accordingly, we have revised the manuscript. We hope the changes have appropriately incorporated your suggestions and are satisfactory. The changes in the revised manuscript are shown in red, and below, we have answered your queries in a point-by-point manner and indicated the exact page and line numbers of the changes.

Reviewer 1

The paper presents the results of experimental in vivo studies on influence of tadalafil on the activation of mTOR in fetal in mice. The Authors emphasized that proposed manuscript continues their study on the identification of molecular mechanism of tadalafil positive effect on fetal growth in an FGR mouse model. The manuscript presents results interesting for readers, however there are some details which need to be explain more specific:

Comment: In the Introduction present more detailed information about significance of mTOR and its signaling targets on fetal growth and development;

Response: Thank you for your insightful suggestion. Accordingly, we have updated the Introduction section. We have added information on the relationship between fetal growth and mTOR signaling. We have also cited relevant and related studies and updated the list of references.

Comment: present more details about chosen dose of studied compounds;

Response: Thank you for pointing this out. We have added the average doses of L-NAME and tadalafil to the Material and Methods section.

Comment: describe the number of animals in each group studied;

Response: The number of animals was added in the Materials and Methods section.

Comment: In case of western blots please add the molecular weight of detected protein bands; explain why in the immunoblots (Figure 3, Figure 4) are visible two bands and which of them represents p4E-BP-1 (Thr 37/46) or p4E-BP-1 (Thr70); please, add the band of unphosphorylated 4E-BP-1 protein, as well as for S6R – as a control of changes resulting by phosphorylation activity; was the level of mTOR protein determined?

Response: Thank you for this helpful comment. Accordingly, we have updated the information on the molecular weights of the proteins in the Figure. The molecular weight of 4E-BP1 is 15–20 kDa, and the band may appear thick or multiple bands in western blot (Reference; https://www.cellsignal.jp/products/primary-antibodies/4e-bp1-antibody/9452).

Unfortunately, we could not detect the unphosphorylated proteins, as we had the antibodies to detect total protein (#9452 and #2217). We have updated the western blot data of total and phosphorylated proteins in Figures 3 and 4. We have also included this information in the methods section. 

Reviewer 2 Report

General comments:

This manuscript consists of 8 numbered pages cited 13 references. The references are properly collected and arranged. In formal aspect it is an appropriate work, and meets the requirements determined by the journal.

Specific comments:

The objective of this manuscript is to provide the key mechanism about the mode of action of tadalafil. These results may support the future clinical studies.

The introduction part is too short and not particular. More references needed to support the background of this study. The study objective is not clear.

The results are more serious, however, the quality of the figures are not satisfactory and the figure legends are also too short and not particular.

The discussion is moderate, but the authors should discuss more references than they did in  he text.

Author Response

Point-by-point responses to the comments of the Reviewers

Dear Reviewer,

Thank you for your efforts to review our manuscript. Thank you very much for your helpful comments and suggestions. Accordingly, we have revised the manuscript. We hope the changes have appropriately incorporated your suggestions and are satisfactory. The changes in the revised manuscript are shown in red, and below, we have answered your queries in a point-by-point manner and indicated the exact page and line numbers of the changes.

Reviewer 2

This manuscript consists of 8 numbered pages cited 13 references. The references are properly collected and arranged. In formal aspect it is an appropriate work, and meets the requirements determined by the journal.

Specific comments:

Comment: The objective of this manuscript is to provide the key mechanism about the mode of action of tadalafil. These results may support the future clinical studies.

Response: Thank you for your insightful suggestion. We have revised the introduction section by highlighting the significance of mTOR signaling in understanding the mechanism of action of tadalafil. We have also highlighted our hypothesis and revised the objectives statement. As suggested, we have added a sentence showing the significance of our study in future clinical studies.

Comment: The introduction part is too short and not particular. More references needed to support the background of this study. The study objective is not clear.

Response: We have added the information on the relationship between fetal growth and mTOR signaling. We have also cited relevant and related studies and updated the list of references. We believe that the addition clarifies the reason for evaluating mTOR signaling and the purpose of the study.

Comment: The results are more serious, however, the quality of the figures are not satisfactory and the figure legends are also too short and not particular.

Response: We have revised the Figure and rewritten the figure legends in detail.

Comment: The discussion is moderate, but the authors should discuss more references than they did in the text.

Response: Thank you for this suggestion. Accordingly, we have revised the discussion section and cited the related and relevant studies.

Round 2

Reviewer 1 Report

The Authors answered my concerns and I suggest the acceptance of manuscript for publication. 

Reviewer 2 Report

It can be accepted